# Epidrugs in the Therapy of Central Nervous System Disorders: A Way to Drive on?

**DOI:** 10.3390/cells12111464

**Published:** 2023-05-24

**Authors:** Marina G. Gladkova, Este Leidmaa, Elmira A. Anderzhanova

**Affiliations:** 1Faculty of Bioengineering and Bioinformatics, Lomonosov Moscow State University, 119234 Moscow, Russia; 2Institute of Molecular Psychiatry, Medical Faculty, University of Bonn, 53127 Bonn, Germany; 3Institute of Biomedicine and Translational Medicine, Department of Physiology, University of Tartu, 50411 Tartu, Estonia; 4School of Medicine, BAU International University Batumi, Batumi 6010, Georgia

**Keywords:** epigenetics, epidrugs, KMTis, DNMTis, HATis, HDACis, BETis, neuroprotection, neuroplasticity, neuroinflammation, neurological and psychiatric disorders, neurodegenerative diseases, lifestyle factors

## Abstract

The polygenic nature of neurological and psychiatric syndromes and the significant impact of environmental factors on the underlying developmental, homeostatic, and neuroplastic mechanisms suggest that an efficient therapy for these disorders should be a complex one. Pharmacological interventions with drugs selectively influencing the epigenetic landscape (epidrugs) allow one to hit multiple targets, therefore, assumably addressing a wide spectrum of genetic and environmental mechanisms of central nervous system (CNS) disorders. The aim of this review is to understand what fundamental pathological mechanisms would be optimal to target with epidrugs in the treatment of neurological or psychiatric complications. To date, the use of histone deacetylases and DNA methyltransferase inhibitors (HDACis and DNMTis) in the clinic is focused on the treatment of neoplasms (mainly of a glial origin) and is based on the cytostatic and cytotoxic actions of these compounds. Preclinical data show that besides this activity, inhibitors of histone deacetylases, DNA methyltransferases, bromodomains, and ten-eleven translocation (TET) proteins impact the expression of neuroimmune inflammation mediators (cytokines and pro-apoptotic factors), neurotrophins (brain-derived neurotropic factor (BDNF) and nerve growth factor (NGF)), ion channels, ionotropic receptors, as well as pathoproteins (β-amyloid, tau protein, and α-synuclein). Based on this profile of activities, epidrugs may be favorable as a treatment for neurodegenerative diseases. For the treatment of neurodevelopmental disorders, drug addiction, as well as anxiety disorders, depression, schizophrenia, and epilepsy, contemporary epidrugs still require further development concerning a tuning of pharmacological effects, reduction in toxicity, and development of efficient treatment protocols. A promising strategy to further clarify the potential targets of epidrugs as therapeutic means to cure neurological and psychiatric syndromes is the profiling of the epigenetic mechanisms, which have evolved upon actions of complex physiological lifestyle factors, such as diet and physical exercise, and which are effective in the management of neurodegenerative diseases and dementia.

## 1. Epigenetics and Approaches to Its Study

The deoxyribonucleic acid (DNA) molecule encoding for proteins is essential for life and is predominantly responsible for immutable traits. The DNA in a cell exists in a strictly structured form that is achieved with the help of histones and architectural proteins. This structural organization defines the differential functional activity of the DNA molecule, which can be transcriptionally inaccessible in dense regions (heterochromatin) or transcriptionally accessible (euchromatin) in loose regions of chromatin (Figure 1).

Epigenetics, on the contrary, gives a living creature the opportunity to adapt to dynamically changing environmental factors, hence, playing an important role in the balance of molecular processes, which, if violated, can lead to various diseases. An epigenetic phenomenon is a change in gene expression that is not mediated by changes in the nucleotide sequence of a DNA molecule. The alterations in DNA accessibility and transcriptional activity are mostly achieved by a few mechanisms: (1) chemical modifications (reversible methylation) of DNA, (2) chemical modifications (e.g., reversible acetylation or methylation) of histone proteins, (3) chromatin remodeling and the spatial three-dimensional (3D) organization of the genome, as well as via action of (4) non-coding ribonucleic acids (ncRNAs) on gene expression. At the cellular and systemic levels, epigenetic mechanisms controlling gene activity influence molecular, biochemical, and biological processes. Epigenetic control over the genome is essential in determining the unique properties of a cell, which develops from a single precursor via specialized gene expression programs.

The upper-mentioned epigenetic functional and structural alterations normally appear in a given organism but may be transmitted to the next generation(s) due to the stabilization of the specific expression programs (with no changes in the genome sequence per se), for instance, via the preservation of the residual methylation after fertilization and via gamete-delivered RNAs [4,5].

Epigenetic mechanisms of all known types can be reliably monitored nowadays. For instance, histone modifications are well recognized as an integrative marker of genome regulation. Chromatin immunoprecipitation (ChIP)-sequencing is commonly used to assess the profile of histone lysine (K) acetylation and methylation. The most validated specific markers of genome suppression are histone H3 methylation at lysine in two positions: H3K9me3 and H3K27me3. In turn, H3K27ac, H3K4me3, and H3K4me1 are validated as genome activation factors. However, to date, a variety of other post-translational histone2-4 modifications, such as arginine acetylation, phosphorylation, SUMOylation, and ubiquitination, are shown to result in changes in genome activity [6].

Table 1 summarizes the contemporary methods of epigenomics, including epigenome next-generation sequencing (NGS), which are used to analyze the epigenetic component of cell activity.

The sensitivity and efficiency of modern analytical methods allow one to assess the epigenetic landscape and reliably identify the promoter, enhancer, as well as associated insulator sequences at the single-cell level [7,8,9]. 

Progress in entire genome chromatin profiling and epigenome-wide association studies (EWASs) (by analogy with a genome-wide association study, GWAS) fostered a surge of data about epigenetic states in various tissues and cells at their normal and pathological functionings. The contemporary challenges of epigenetic data management are the (1) development of open-access databases; (2) standardisation of experimental methods; (3) setting of standards for the analysis and integration of EWAS and GWAS data; (4) annotation of epigenetic markers at different levels of biological complexity; and (5) selection of a significant set of epigenetic markers in the context of a disease or treatment [8,10,11,12,13,14].

Aside from the general constraints related to big data management, there are a few additional obstacles to understanding the role of epigenetic mechanisms in the pathogenesis of neurological and psychiatric diseases. Although a big pool of information on changes in the epigenome is amassed to date, we are still limited in detailed aspects of the dynamics of epigenetic landscapes in different neuronal cells in various brain regions. Similarly, a systematic search for reliable peripheral epigenetic markers of central nervous system (CNS) disorders has still not been accomplished. The rationale for using peripheral epigenetic markers, in addition to their easy accessibility in humans, is the involvement of peripheral systems (namely, the immune system, circulating stress hormones, myokines, and the microbiome) in the normal maintenance of brain functions and the generation of pathological endophenotypes. One of the possible platforms for finding significant correlations can be the concept of a so-called epigenetic “clock”, which assumes the age-related accumulation of methyl groups on a DNA molecule, allowing one to determine the biological age of cells, tissues, organs, or organisms. The best-validated model is Steve Horvath’s epigenetic “clock”, comprising 353 epigenetic markers of the human genome [15,16].

## 2. Epigenetic Therapy of Diseases of the CNS

### 2.1. Epigenetically Active Drugs (Epidrugs)

At present, the field of epidrug pharmacology is developing to serve mainly the needs of cancer therapy. However, epigenetically active substances, being, by default, multitarget and multifunctional, would represent the cluster of drugs of which specific and unspecific effects are potentially even of higher interest with respect to other nosologies or syndromes [17]. The known compounds that are able to interfere with epigenetic mechanisms are represented by drugs with a primary epigenetic activity or by those that possess epigenetic effects as a side phenomenon. Table 2 lists existing epigenetically active substances (epidrugs) in accordance with the biochemical mechanism [18,19,20]. To date, the compounds of three types have been studied the most: (1) inhibitors of DNA methyltransferases (DNMTis); (2) inhibitors of histone deacetylases (HATis); and (3) inhibitors of DNA demethylases (TETis). Several compounds with primary epigenetic activity have been approved as anticancer therapy by the EMA (European Medicines Agency) and/or the FDA (Food and Drug Administration) of the USA, as well as the FDA of the People’s Republic of China (e.g., chidamide).

**Table 2 cells-12-01464-t002:** Clusters of epidrugs and compounds exerting epigenetic activities. Approved anticancer compounds with primary epigenetic activity are in bold; subclasses of inhibitors are italicized.

Epigenetic Modification/ Pathway (Figure 2)	Target Domain and Mechanism of Action	Abbreviation	Examples of Approved Compounds	Primary Target Pathology
DNA methylation	DNA methyltransferase inhibitors	DNMTis	5-azacytidine, decitabine [21], hydralazine [22], and procainamide [23]	myeloid neoplasms, malaria, and cardiovascular diseases
DNA demethylation	Ten-eleven translocation (TET) proteins inhibitors	TETis	-	
Histone acetylation	Histone acetyltransferase inhibitors	HATis	anacardic acid, curcumin, garcinol, catechin, and thiazole + bistubstrate inhibitors [24]	antimicrobial therapy, anti-inflammatory therapy, and cancer
Histone deacetylation	Histone deacetylases inhibitors	HDACis	**vorinostat**, **panobinostat** (withdrawn by FDA in 2022), **romidepsin** (withdrawn by FDA in 2021), **belinostat**, **chidamide (tucidinostat)** [25], entinostat (granted breakthrough therapy status by the FDA in 2013), valproic acid, magnesium salt of valproic acid, phenylbutyric acid [2], nicotinamide (affirmed as GRAS (Generally Recognized as Safe) by the FDA in 2005 as a direct human food ingredient) [26], and carbamazepine [27]	lymphomas, myeloid neoplasms, other types of cancer, epilepsy, and dietary supplement
Histone methylation	Histone methyltransferase inhibitors	HMTis	-	
	*Lysine-specific histone* *methyltransferases* *inhibitors*	*HKMTis*	*phenelzine* [28], *tranylcypromine* [29], and *tazemetostat* [30]	*major depression*, *anxiety*, *and epithelioid sarcoma*
Histone demethylation	Histone demethylase inhibitors	HDMis	deferiprone [31] and deferasirox [32]	treatment of iron overload in thalassemia and major or long-term blood transfusions
	*Lysine-specific histone* *demethylases inhibitors*	*LSDis/* *KDMis*	-	
Protein arginine methylation	Protein arginine methyltransferase inhibitors	PRMTis	-	
Protein arginine citrullination (deamination)	Protein arginine deiminase inhibitors	PADis	streptomycin and methotrexate [33]	antimicrobial therapy, chemotherapy agent, and immune system suppressant
Phosphorylation	Histone kinase inhibitors	-	ruxolitinib, pacritinib, pazopanib, vandetanib, lapatinib, and erlotinib [25]	myelofibrosis, atopic dermatitis, vitiligo, renal cell carcinoma, soft tissue sarcoma, medullary thyroid cancer, breast cancer, non-small cell lung cancer, and pancreatic cancer
Ubiquitination/ deubiquitination	Inhibitors of ubiquitin signaling modulators (proteasome, target E1, E3, or DUB modulators)	-	bortezomib, carfizomib, ixazomib, thalidomide, lenalidomide, and pomalidomide [34]	multiple myeloma, mantel cell lymphoma, and myelodysplastic syndromes
Poly(ADP-ribosyl)ation (PARylation)	Poly(ADP-ribose) polymerase (PARP1) inhibitors	PARPis	olaparib, niraparib, rucaparib, and talazoparib [35]	different types of cancer
Others	Inhibitors of proteins binding to acetylated histones	PAHis	-	
	*Bromodomain and Extra terminal* *motif proteins (BETs) inhibitors*	*BETis*	*dinaciclib (granted orphan drug* *status by the FDA in 2011)* [36]	*different types of cancer*
	Inhibitors of proteins binding to methylated histones	PMHis	-	
	Regulation by non-coding RNA	ncRNAs	patisiran, givosiran, and pegatanib [37]	familial amyloid polyneuropathy, hepatic porphyria, and macular degeneration

Limitations to use the epidrugs are related to (1) their chemical instability; (2) ubiquitous activity of DNMTis accompanied by the development of side effects; (3) unspecific acetylation, deacetylation, and methylation of non-histone proteins resulting in undesirable side effects (such as fatigue and dysfunction of the gastrointestinal tract); and (4) cytotoxic effects and significant inhibition of hematopoiesis [38].

**Figure 2 cells-12-01464-f002:**
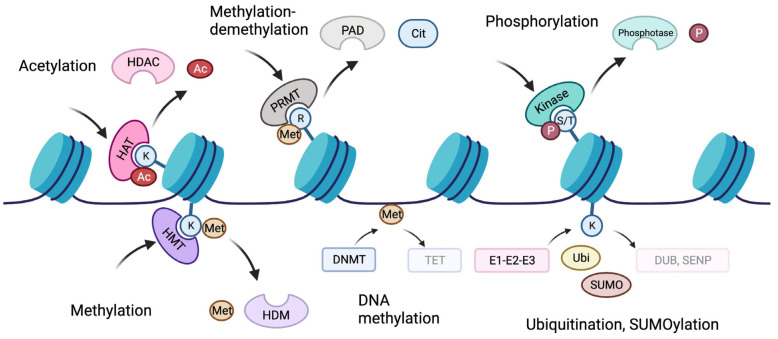
Epigenetic mechanisms. A DNA strand can be dynamically modified, mostly by adding methyl groups. Cytosine residues in DNA can be methylated by DNA methyltransferases (DNMTs) to 5-methylcytosine (5mC) (found primarily at CpG sites), which is oxidized to 5-hydroxymethylcytosine (5hmC) by ten-eleven translocation (TET) proteins. The 5hmC can be further oxidized by TET proteins to become 5-formylcytosine (5fC) and then 5-carboxylcytosine (5caC), or deaminated by activation-induced cytidine deaminase (AID) or APOBECs. In turn, accessibility can be controlled via histones. The parts of histone molecules protruding from nucleosomes can be modified by special “writer” proteins, adding different chemical groups (methyl, acetyl, phosphate, crotonyl, citrate, and serotonyl) or small proteins (ubiquitin and SUMO). Special “eraser” proteins remove marks from histones. Other proteins, called “readers”, recognize and interact with the certain marks of histone “tails” (N-terminus) to mediate specific transcription. Sometimes only one modification can be enough to regulate processes on chromatin. After the discovery of the first histone modifications, David Allis and Brian Strahl put forward the “histone code” hypothesis: a combination of histone marks reflects a code that is read by other proteins and determines the molecular processes at the site [39]. However, it seems that the real picture is more complicated, and not only histone marks control molecular processes on chromatin. Abbreviations: Cit = citrulline; DNMT = DNA methyltransferase; DUB = deubiquitylating enzyme; E1 = ubiquitin-activating enzyme; E2 = ubiquitin-conjugating enzyme; E3 = ubiquitin ligase; HAT = histone acetyltransferase; HDAC = histone deacetylase; HDM = histone demethylase; HMT = histone methyltransferase; PAD = peptidyl arginine deiminase; P = phosphate group; PRMT = protein arginine methyltransferase; R = arginine histone residue; S = serine histone residue; SENP = sentrin-specific protease; T = threonine histone residue; TET = TET protein; K = lysine histone residue; and Met = methyl group. Created with BioRender.com (accessed on 2 August 2022).

### 2.2. Significance and Requirements for Epigenetic Therapy for CNS Diseases

The increase in life expectancy, changes in lifestyle, and informational load resulted in the emergence of new pathogenic factors (such as aging and stress) and the decrease in significance of old pathogenic factors (such as infections and nutritional deficiencies). The profiling of CNS diseases in the world population shows a growing dominance of neurodegenerative disorders and psychophysiological stress-associated disorders [40,41,42]. Such a bias in the profile of pathologies defines new therapeutic goals and further stresses the fact that the contemporary therapeutic pharmacological approaches are not sufficiently effective. This situation challenges scientific society and the pharmaceutical industry and, among others, increases an interest in epidrugs as a means to treat CNS disorders.

Table 3 provides an overview of epidrugs showing activity to treat pathologies of the CNS in preclinical (mostly) and clinical studies.

While we already know a number of compounds with primary epigenetic activity, their rapid transfer from preclinical studies into the therapy of the CNS is hindered by some flaws and difficulties.

A list of the most crucial demands on epidrugs includes: Permeability of the blood–brain barrier (BBB);Regionality of action (for example, mainly in cortical regions);Differential specificity of action at the level of neuronal or glial populations (might be crucial for a selective targeting of neuroinflammation [63]);High selectivity of compounds to their molecular targets (at the level of protein isoforms);If the drug is multitargeted, then its molecular action has to be highly balanced with no or few side effects;Absence of cytotoxic effects on cells with a non-cancer phenotype;Lack of systemic immunosuppressive and hematopoiesis-inhibiting effects;Possibility of a differential therapy according to the phase of the disease (for example, conducting early postnatal treatment of pathologies associated with genetic disorders).

Based on the requirements and recent promising findings on the mechanisms of CNS disorders, the following cues for epidrug research and development can be delineated: (1) the search for compounds with good BBB permeability among selective inhibitors of TET proteins and DNA methyltransferases; (2) the search for compounds with high potency to selectively regulate neuroimmune inflammation and neurotrophic mechanisms; (3) the epigenetic activity profiling and assessing of the possible “off-label” use of compounds with canonical effects, such as anticonvulsant and antidepressant agents or metabolites; and (4) the employment of CRISPRon and CRISPRoff gene-editing technologies, as well as more advanced CRISPR/Cas system versions (such as Prime editing). The last is of particular interest because it enables cell- and locus-specific reactivation or repression of the transcription of a single gene or a combination of genes (multiplexed gene targeting) under a variety of promoters [64]. However, the clinical use of the transgenic approaches is still limited due to the problem of targeted delivery of compounds and genetic constructs to the human brain. The approach also needs to be more careful in addressing possible off-target effects.

When determining the strategies of therapeutic intervention in the treatment of CNS disorders, one should take into account the ability of complex physiological factors (lifestyle factors), such as physical activity, dietary factors, and fasting, as well as cognitive activity, to modulate the epigenetic landscape. Being ministers of beneficiary action (e.g., antidepressant and pro-cognitive), these factors do not lead to the development of unwanted side effects. Therefore, their epigenetic sequels may be considered as a prototype to explore the potential targets for specific epigenetic therapy.

## 3. Biological Targets of Epidrugs

The very nature of epidrugs points to their complexity and diversity of effects. On the cellular level, epidrugs influence fundamental processes, such as central metabolism, autophagy, cell survival, and the reprogramming of pluripotent or cancer cells. On the tissue level, the effects of epidrugs appear as an activation of neuroprotective and neuroplastic mechanisms as well as of the modulation of neuroinflammation, which involves the intercellular interaction and reorganization of the extracellular matrix. At the next level of complexity, epidrugs target particular endophenotypes and syndromes. A final achievement would be the correction of pathologies on the organismal level. 

Figure 3 summarizes the most-studied effects of epidrugs, which are grouped according to their biological complexity levels.

To make an inventory of the available substances with epigenetic activity and to overview the contemporary tendencies in preclinical and clinical studies of their applicability for CNS treatments, we scrutinized respective publications based on the following logic.

Substances belonging to the continuously growing cluster of epidrugs were, first, stratified based on the main known biochemical/epigenetic mechanisms.

Second, epidrugs from each cluster with defined biochemical mechanisms were looked for in reports on the results of preclinical or clinical studies of therapeutic effects. Then, they were clustered in groups, which were formed with respect to the pathologies and pathology-related mechanisms that should be targeted (cytostatic action, neuroprotection, or neuroregeneration, treatment of neurological neurodegenerative disorders, and treatment of psychopathological syndromes).

We lined up the results of our search with respect to the number of published studies, which reflect the current interest in the respective epidrugs as therapeutic means for treating neurological and psychiatric disorders.

We also stressed the role of complex epigenetic regulators in the therapy of certain diseases, assuming their actual benefits.

### 3.1. Epigenetic Therapy for Brain Cancer

The strategy for treating cancer with epigenetically active compounds implies their use as adjuvant therapy [65]. To this end, the cytotoxic and/or cytostatic effects, as well as the ability of epidrugs to activate immune mechanisms, are exploited [66].

Nonselective HDACis are currently undergoing phase II clinical trials as a component of complex therapy for brain cancer [67,68,69,70]. The study on selective targets of HDACis singled out HDAC6. This enzyme shows a relatively high level of expression in the brain, which may be a prerequisite for the high efficacy of specific inhibitors. Moreover, selective HDAC6is, MPT0B291 and tubacin, show cytotoxic and cytostatic activity against human and rat glioma cells, but not normal astrocytes, both in vitro and in vivo [71,72]. The efficiency of the HDAC6 inhibitor JOC1 in the treatment of multiform glioblastoma was also demonstrated in clinical studies [73].

The spectrum of pharmacological activity of nonselective HDACis may be the determinant of a complex pharmacotherapy for brain cancer. For example, panobinostat, vorinostat, and romidepsin inhibit glycolysis in cancer cells by enhancing the oxidative metabolism (potentiating the expression of peroxisome proliferator-activated receptor gamma coactivator 1-alpha (PGC1α) and peroxisome proliferator-activated receptor beta (PPARδ)). Since the cancer oxidative metabolism is mainly driven by fatty acid oxidation, a combination of fatty acid oxidation inhibitors and HDACis might be effective in cancer treatment, as it was proved in a patient-derived xenograft model of glioblastoma in mice [74].

BET inhibitors show cytotoxic activity against glioblastoma cells in vitro [75]. The synergistic cytotoxic effect of the BET inhibitor JQ-1 and the transcriptional activator CREB-binding protein (CBP), which recruiats bromodomain (BRD) proteins, showed to be effective as a therapy for diffuse intrinsic pontine glioma [76]. JQ-1 inhibits the growth of glioma implanted in mice when a transfer across the BBB is ensured using PEGylated nanoparticles [77].

DNMTis are also studied as adjuvant components in the treatment of brain cancer [78,79].

The experimental evidence and results of clinical trials accumulated to date strongly suggest that epidrugs may be effective in the treatment of brain neoplasms.

### 3.2. Neuroprotective and Pro-Regenerative Effects of Compounds with Epigenetic Activity

Cell death, which involves a wide spectrum of scenarios, can, besides being a part of the pathogenesis of neurodegenerative disorders, be assumed to be a leading component of degeneration in brain and spinal cord injuries, brain hemorrhage, ischemic stroke, chronic insufficiency of cerebral circulation, and toxicity-related damage (including that related to toxic metabolic syndromes).

HDACis showed neuroprotective effects in models of traumatic brain injury. Among the few mechanisms involved, the decrease in the neuroinflammatory response is particularly noteworthy [80]. Suppression of neuroinflammation is determined by an epidrug-evoked decline in the secretion of proinflammatory cytokines, suppression of glial cell proliferation, and changes in glial morphology [81,82,83,84,85]. This action of epidrugs is also associated with the increased expression of neurotrophic factors BDNF and NGF [86,87]. Additionally, several studies showed that the components of intracellular signaling, which increase cell survival (for example, the AKT-GSK3ß kinase cascade), are affected by HDACis [88,89]. Notably, the involvement of the PI3K-AKT-GSK3ß signaling pathway in the mechanisms of the neuroprotective effects of epidrugs emphasizes a synergism of HDACis and the activity of “conventional” drugs targeting the insulin- and serotonin-dependent intracellular signaling pathways [90]. Insufficiency of this cooperation may be an important determinant of resistance to tricyclic antidepressants and SSRIs [91,92,93].

The neuroprotective effect of nonspecific HDACis in models of ischemia, oxidative stress, and glutamate neurotoxicity is associated with the transcriptional suppression of pro-apoptotic factors, such as p75(NTR)-dependent caspase-3 and ubiquitin-conjugating enzyme E2N Ube2n [94,95,96]. Additionally, nonspecific HDACi valproic acid affects mitochondrial biogenesis [97] and reduces the oxidative stress caused by psychophysiological stress or global cerebral ischemia [98]. HDACis also show suppression of glial differentiation and activity and potentiation of neuron differentiation from progenitor cells during adult neurogenesis [99,100]. However, the prospects for the use of HDACis for nerve tissue bioengineering, with the help of pluripotent cell reprogramming technologies, is not entirely clear yet [101,102]. A limitation to the use of nonspecific epidrugs is related to a possible acetylation/deacetylation imbalance. Garcinol, which combines the properties of an HAT p300 inhibitor and an HDAC11 inhibitor, potentiates toxic effects in the 1-methyl-4-phenylpyridinium (MPP^+^) toxicity model [103].

Several observations further illustrate the importance of selective modulators of histone acetylation and deacetylation in realization of a neuroprotective effect [104,105]. The therapeutic effect of exifone, a selective activator of HDAC1, was previously attributed to its scavenger properties [106]. However, exifone-driven neuroprotection in the model of oxidative stress associated with the accumulation of 8-oxo-2-deoxyguanosine rather depends on the activation of HDAC1 and the reduction in H4K12 acetylation levels in neurons. This protective effect of exifone was not observed in astrocytes, which implies a cell-type-specific action [55,107]. Advantages of the selective inhibition of histone acetylation were also demonstrated in the model of peripheral nerve regeneration [108]. The effect of a nonspecific HDACi MS-275 in the axonotomy regeneration model was weaker than the effect of the histone acetyltransferase p300/CBP-associated factor (PCAF) overexpression [108,109]. 

A nonspecific DNMTi, 5-aza-deoxycytidine, potentiated MPP+ toxicity in dopaminergic neurons [43]. Another unspecific DNMT inhibitor zebularine, a nucleoside analogue of cytidine, increased the accumulation of β-amyloid in N2a mouse neuroblastoma cells in vitro [44]. In contrast, the DNMT1i RG108 suppressed apoptosis in motor neurons, providing neuroprotective effects [46]. Although the use of nonselective DNMTis does not appear to be a promising way for regulating neuronal function, the isoform-selective DNMT1 and DNMT3a/b inhibitors may nevertheless be more suitable [110,111].

BET inhibitors, especially those selective for BRD2/4 proteins, exhibit a characteristic suppression of the expression levels of pro-apoptotic factors and cytokines regulating inflammation (tumor necrosis factor alpha (TNF-α), C-X-C motif ligands 1 and 10 (CXCL1; CXCL10), and C-C motif ligand 2 (CCL2)), as well as matrix metalloproteinase 9 (MMP9), the extracellular cleaving enzyme involved in the maintenance of the extracellular matrix and membrane receptors [112,113,114,115,116].

Overall, known epidrugs facilitate the neurotrophic and neuroplastic processes, inhibit the mechanisms of programmed cell death, and suppress neuroinflammation. These features prompt researchers to look at the applicability of certain epidrugs as neuroprotectors and stimulators of neuronal regeneration but also suggest the domain of neurodegenerative disorders as a potential area of applicability.

### 3.3. Specific Therapy for Neurodegenerative Diseases

Epidrugs potentially can be used in the treatment of neurodegenerative disorders because they activate neuroprotective mechanisms, suppress neuroinflammation, and evoke the mechanisms of synaptogenesis and neuroplasticity [117]. Considering this, epidrugs affect specific aspects of the development and pathogenesis of neurodegenerative diseases by modulating expression and post-translational modifications of functional proteins as well as pathoproteins.

The most prevalent neurodegenerative diseases, such as Huntington’s disease [118], Parkinson’s disease [119,120], Alzheimer’s disease [121,122,123], and amyotrophic lateral sclerosis [124], are characterized by a dysregulation in histone acetylation in the brain. However, it is worth mentioning that in the case of Parkinson’s disease, the genome-wide H3K27ac hyperacetylation was found in the cortex [120], not only in the midbrain [119].

Despite the differential regulation of histones in neurodegenerative disorders, it is assumed that the use of HDACis can compensate for these pathological changes [125,126] via a complex mechanism including downregulation of pathoprotein production and inflammatory cytokines [127,128]. Besides targeting specific pathogenetic mechanisms, HDACis may influence the efficiency of learning and memory [129], therefore compensating for pathological phenotypes.

HDAC1/2 inhibitor BG45 enhances the level of synaptic proteins PSD95, spinophilin, and synaptophysin in the model of toxic β-amyloid expression in vitro [130]. In turn, selective HDACis BG45 and MGCD0103 (mocetinostat) not only reduce the neurotoxic effect of β-amyloid but also decrease its production [130,131]. At the same time, HDACis MS275 and CI994 increase the expression of apolipoprotein E4 [132], a risk factor gene associated with Alzheimer’s disease [133]. HDAC4/5 inhibitors and nonspecific HDACis suppress α-synuclein production Parkinson’s disease [134,135]. 

The use of HDACi LBH589 (panobinostat) in the early postnatal period in the genetic model of Huntington’s disease leads to a significant improvement in behavioral and neurochemical endophenotypes [136], which points to interference with the neurodevelopmental mechanisms of this disorder. Nonspecific HDACis sodium butyrate and trichostatin A normalize mitochondrial activity and, therefore, prevent impairments in motor learning and coordination in YAC128 transgenic mice (full-length *HTT* expression) [137]. The HDAC3 inhibitor RGFP966 activates the expression of genes characteristic of neuroplastic processes in the hippocampus of Huntington’s disease gene homolog *(Hdh)* knockout mice (Hdh^Q111^-KO), which is accompanied by an improvement in cognitive functions [138]. 

Besides interference with histone acetylation, DNA methylation is also considered as a target for epigenetic therapy of neurodegenerative disorders [139]. A differential profile of DNA methylation is shown in the brain of patients with Parkinson’s [140] and Alzheimer’s [141] diseases. The use of lysine demethylase inhibitors (LSDis or KDMis) appears to be a promising strategy to prevent neurodegeneration. ORY-2001 (vafidemstat), combining properties of a KDM1A inhibitor and a monoamine oxidase B (MAO B) inhibitor [142], was reported to reduce agitation and aggression, as well as cause a rise in the total Neuropsychiatric Index and measures of caregiver burden in an open-label study in 12 people with Alzheimer’s disease (a six-month course of 1.2 mg of the drug daily) [143]. A clinical trial on the safety of this compound in the treatment of Alzheimer’s disease was recently completed [143].

The BETi JQ-1 affects pathological behavioral and molecular biological endophenotypes in a model of Alzheimer’s disease in mice. JQ-1 reduces Tau protein phosphorylation, an expression of pro-inflammatory factor genes (*Il-1b*, *Il-6*, *Tnfa*, *Ccl2*, *Nos2*, and *Ptgs2*) in 3xTg mice [144], prevents a decrease in cognitive functions, and restores the physiologically relevant expression of genes (e.g., ion channels and DNA repair) in the hippocampus in APP/PS1-21 mice [145]. Apabetalone, a compound with BET inhibitory activity, improves cognitive performance in people over 70 years of age. However, this may be due to its beneficial effects on the cardiovascular system [146]. 

However, existing HDACis and BETis have pronounced immunosuppressive and cytotoxic effects, which may be an obstacle to their use for CNS disorders, which require long-term treatment. The presence of side effects motivates a search for selectively acting compounds and/or routes of targeted drug delivery or drugs with local action. In this regard, a novel hybrid technology that exploits the capabilities of the CRISPR/dCas (dead Cas) complex, and an epigenetic enzyme recruited by it, is of increasing interest. The expression of such constructs allows a locus-specific reactivation (CRISPRon) or repression (CRISPRoff) of DNA transcription [64,147,148]. However, this promising approach requires significant optimization to achieve a high cell-specificity of the construct’s expression in humans. It should be mentioned that in addition to the use of viral vectors with BBB permeability, the possibility of using PEGylated immunoliposomes demonstrating good BBB permeability and high biocompatibility is being considered [149].

### 3.4. Epigenetic Therapy of Psychopathological Syndromes

The design or selection of epidrugs for the treatment of psychiatric disorders is determined by the characteristic biological features of psychiatric syndromes: (1) high polygenicity with small effect sizes of individual risk variants; (2) strong influence of external and internal environmental factors on the development and dynamics of psychopathological endophenotypes; and (3) overlap of pathogenetic mechanisms between psychopathological syndromes and neurodegenerative diseases (neuroinflammation, maladaptive neuroplasticity, and neurodegeneration).

It is also important to note that known psychoactive pharmacological substances are being proven to have epigenetic mechanisms of action. In turn, it is not surprising that many epigenetic drugs mediate psychical activity because they affect neuroplasticity, which is essential to maintaining functional homeostasis and adaptive changes in the brain.

#### 3.4.1. Possibility for Off-Label Use of Substances with Known Psychopharmacological Activity

Antidepressants, antipsychotics, and drugs with antiepileptic activity are constantly confirmed to have epigenetic effects. These epigenetic activities may be the main determinants of the clinical effectiveness of these drugs, instead of their initially proposed modes of action. For instance, imipramine was introduced for depression treatment as early as 1957. As shown more recently, its antidepressant effect in a chronic stress mouse model is mediated by an increase in H3 acetylation in the *Bdnf* gene promoter and BDNF expression [150]. Valproic acid has been in use since 1962. Besides acting as an antiepileptic drug, it was proposed as an add-on therapeutic means to cure schizophrenia and bipolar disorder [151,152]. The administration of valproic acid is accompanied by an increase in the levels of H3 and H4 acetylation in the cortex and hippocampus. This regionality of epigenetic changes seems to be necessary for the effects of valproic acid in the treatment of panic attacks and anxiety disorders in humans, conditions with insufficient top–down control of emotions and general excitability [53,54]. Together with the revitalization of the known drugs and their off-label use, identifying their epigenetic effects may provide a new avenue for developing derivatives with primary epigenetic activity [150,153,154,155,156].

#### 3.4.2. Drugs with Primary Epigenetic Activity

Effects of HDACis and BETis in the CNS are well known to be associated with an increase in the transcription of genes regulating neurogenesis, synaptogenesis, and neuroplasticity, a decrease in cytokine production, and the suppression of microglial activity. An increasing number of evidence indicates the effects of HDACis and BETis on the expression of neurotransmitter and neurohormone receptors and transporters (e.g., dopamine, serotonin, gamma-aminobutyric acid, glutamate, and corticosterone) [157,158,159,160,161]. The regulation of the expression of conventional and physiologically significant targets of psychoactive drugs further supports the idea of the prospective use of epigenetically active compounds as therapeutic agents for psychopathological conditions.

Furthermore, DNMTis hold promise for their use in treatments of stress-associated psychiatric diseases. Global and locus-specific DNA methylation reflects the active response of neuronal tissue to stress [162] and/or its ability to cope with stress (stress resilience) [150]. The direct [163] and indirect [154] reduction in global methylation leads to an antidepressant effect [164]. Psychophysiological stress is one of the leading pathogenetic factors of depression and post-traumatic stress disorder (PTSD) [165,166]. Increased methylation of the exon 1F of the NR3C1 glucocorticoid receptor is associated with higher hypothalamic–pituitary–adrenal (HPA) axis activity, increased production of cortisol, as well as pathological changes in the behavior of patients with depression and those who have suffered mental trauma in childhood. This may lead to a decrease in the expression of glucocorticoid receptors and impaired activity of the stress hormone axis [154,167,168,169]. The HPA axis can be one of the targets of epigenetic therapy for alleviating stress-related psychopathological syndromes.

Notably, DNMTis are expressed at different levels in various brain regions and are usually present in neurons and not glial cells [110]. Moreover, DNMT-mediated epigenetic regulation in neurons is isoform-specific. Thus, the expression of DNMT3A2 increases in response to stable synaptic activity [170], indicating the dependence of DNA methylation on the excitability of neurons. The effect of DNMT inhibitors on memory and learning appears to be centered on the phases of memory formation (learning, consolidation, and reconsolidation) [171,172,173,174,175]; however, these effects are not fully understood.

The BETi JQ-1, on the other hand, shows contradictory effects in mouse models of PTSD. It reduces the extinction of traumatic memory [176], decreases traumatic memory [177], or reduces the recognition of a familiar object during the object recognition tests assessing memory [60]. These observations indicate the dependence of neuroplastic processes on the activity of BRD1-4 proteins. However, the mechanism behind BETi effects during different phases of memory formation remains unclear.

### 3.5. Epigenetic Therapy for Developmental Disorders, Drug Addiction, Epilepsy, and Pain Disorders

#### 3.5.1. Developmental Disorders

The role of epigenetic mechanisms in developmental genetic disorders has been widely acknowledged [178]. The efficiency of epidrugs in preclinical studies for these disorders is, therefore, not surprising. Thus, both the selective HDAC6 inhibitor SW-100 in the model of Martin–Bell syndrome (fragile X syndrome) [179] and sodium salt of valproic acid in the model of Angelman syndrome [180] prevented the development of the pathological phenotype (impairment of memory and learning, social behavior, and motor functions).

The BETi JQ-1 showed to be beneficial in the model of Rett syndrome [181]. The BET inhibitor I-BET858 altered the expression of genes belonging to the annotation clusters of “neuroplasticity” and “synaptogenesis” in a mouse model of autism. Interestingly, among all the clusters analyzed, only the expression of genes associated with Wnt signaling changed both in the cases of acute and chronic administration of the substance [182].

#### 3.5.2. Drug Addiction

Changes in the epigenome have been demonstrated during the development of addiction to alcohol, nicotine, cocaine, amphetamine, cannabinoids, and opiates [183,184,185,186,187,188,189,190]. Notably, these changes are cell-type-specific; the profiles of epigenetic markers in neurons and astrocytes in response to psychostimulants and opiates differ [191].

Preclinical test results indicate the potential for using HDACis to treat drug addiction [192,193,194,195,196,197]. However, an increase in the dependence rate upon administration of HDACis was also noted, which can result from the interaction of the epidrug activity and the phase of addiction development [198]. 

The potential of BETis in the treatment of drug addiction is generally poorly studied, although the possibility for their use in the treatment of cocaine addiction has been discussed [194].

Neuroplastic changes involved in forming addiction to different drugs seem to be an attainable target for epigenetic therapy. However, since the answer to whether treatment should preferentially affect neuroplasticity in a selected neuronal population is “yes” [199], any success seems to depend on progress in the targeted delivery of epigenetic drugs.

It can also be stressed here that epigenome profiling is crucial to determine the mechanisms leading to sensitivity, resistance, and tolerance to pharmacological drugs [184,200].

#### 3.5.3. Pain Syndromes and Epilepsy

The effects of epidrugs are mainly rapid. At the same time, they are fundamentally reversible. That makes the therapy of epilepsy and pain syndromes another potential field of epidrug application [201]. Indeed, the BET inhibitor JQ-1 reduces seizure activity in the pentylenetetrazole seizure model [60]. The analysis of clusters of enrichment of the full-genomic effect of JQ-1 indicates the significant changes in ionotropic receptors [202]. Changes in the expression of ion channels or other proteins in peripheral nerves, e.g., the mitochondrial transmembrane protein FUNDC1 [203], may underlie the antinociceptive effects of drugs with epigenetic activity [204,205,206].

### 3.6. Lifestyle Factors in Epigenome Modulation in the Treatment of Neurodegenerative Diseases and Psychopathological Conditions

Epigenetics is likely to play a major role in the interaction between the environment (both physical and social) and gene expression. The term “lifestyle” is defined as a complex of modifiable habits and a typical way of living for an individual. It includes factors such as diet, behavior, stress, physical and cognitive activity, working habits, smoking, and alcohol consumption, all of which are shown to alter epigenetic landscapes [207]. Moreover, the measurement of DNA methylation patterns allows to discriminate between individuals with a healthy versus unhealthy lifestyle, quantified by assessing diet, physical activity, and smoking and alcohol intake by individual [208]. A strategy that induces a complex therapeutic effect on the epigenome could thus consider the modulatory influence of lifestyle factors: adherence to sports and a healthy diet, cognitive activity, and the frequency of psychological stress. On the other hand, the specific constellation of epigenetic mechanisms that mediate the action of lifestyle factors may be a reference in the search for more specific epidrugs with similar features. 

It is worth mentioning that due to the systemic effect that specific lifestyle interventions and epidrugs themselves have on metabolism and the immune system, their overall positive impact on brain function and homeostasis of nervous tissue can be, at least in part, a secondary phenomenon.

#### 3.6.1. Inflammation as a Proxy for Epigenetic Changes Evoked by Systemic Lifestyle Factors

Both lifestyle factors and systemic neuroinflammation have been linked to several pathological conditions, including psychiatric and neurodegenerative disorders [209,210,211]. Inflammation is also elevated in obesity, a condition particularly amenable to lifestyle changes [212,213]. Moreover, psychiatric disorders and obesity are shown to co-occur [214]. In obesity, expanding adipose tissue secretes proinflammatory adipokines [215], such as interleukin-6 (IL-6) and TNF-a, which can lead to inflammation and metabolic dysfunction associated with obesity [216]. Interestingly, both IL-6 and TNF-a are also elevated in neurodegenerative and psychiatric disorders [217,218,219,220,221,222,223]. It is thus possible that obesity and psychiatric conditions further potentiate each other, or perhaps, one could lead to another through the mediation of inflammatory changes.

Chronic low-grade inflammation induces a multitude of epigenetic changes. Meta-analysis of EWASs of serum C-reactive protein (CRP), a sensitive marker for low-grade inflammation that is also a proxy for IL-6 levels, found differential methylation at 218 CpG sites to be associated with CRPs [224]. Authors propose that these genetic loci may underlie inflammation and serve as targets for the development of novel therapeutic interventions for inflammation. The more recent meta-analysis found 1511 independent differentially methylated loci associated with CRPs, mostly situated in TF binding sites and genomic enhancer regions [225]. It is also suggested that most of the identified differentially methylated genes are hypomethylated in inflammatory processes [226]. 

Importantly, increased systemic inflammation caused by an energy-dense Western diet is reversible by a change in diet, as switching back to normal food rescues the inflammatory profile in mice susceptible to atherosclerosis [227]. A reduction in body weight might thus be helpful in the management of psychiatric conditions via reduced inflammation and corresponding epigenetic changes. Nevertheless, Western diet-induced transcriptomic and epigenomic reprogramming of myeloid progenitor cells led to their increased proliferation and enhanced innate immune responses, which were not fully reversible after the change to a healthy diet [227]. 

Furthermore, obesity also increases oxidative stress through different mechanisms, including hyperleptinemia, chronic inflammation, low antioxidant defense, and postprandial reactive oxygen species (ROS) generation [228]. ROS and oxidative stress induce a reduction in the activity of DNMTis, which can lead to global hypomethylation. However, by decreasing TET activity, increasing DNMT expression via hypoxia-inducible factor 1 (HIF1), or recruiting DNMT/SIRT1 (HDAC)-containing complexes to H2O2-induced DNA double-strand breaks, ROS can also result in local hypermethylation. Increased levels of ROS have also been widely associated with increased histone acetylation [229].

#### 3.6.2. Dietary Factors Trigger Epigenetic Mechanisms

Dietary change is presumably the most obvious lifestyle intervention to tackle obesity or metabolic dysfunction, and thus modulates systemic inflammation, oxidative stress, and potentially neuroinflammation. At the moment, the dietary factors that have epigenetic effects can be grouped into three classes: single nutrients, particularly vitamins and polyphenols [230]; microRNA from food, which can be safely delivered to organisms and is incorporated into extracellular vesicles [231]; and caloric excess or restriction [232].

The involvement of central metabolites in the regulation of the activity of epigenetic enzymes provides a mechanistic link between dietary and calorie intake and changes in the epigenetic landscape [233]. For example, ATP is required for activation of chromatin-remodeling complexes [234], acetyl-CoA is essential for histone acetylation [235], and NAD+ is a co-factor for sirtuin-type HDACis [236].

In turn, calorie restriction is an appreciated longevity factor, and mechanisms behind its action are mostly found to be linked to reduced nutrient signaling [237]. Fasting or a low-calorie diet drives changes in metabolism and the activation of autophagy and also initiates a complex shift in the epigenome [238,239,240]. Calorie restriction is also accompanied by the augmentation of neuroprotective mechanisms [241,242,243]. However, it is not only high body weight but also body fat distribution that affects the epigenome. Lower DNA methylation age acceleration distances correlate with waist-to-hip ratios in participants with healthy lifestyles [208].

Neurodevelopmental factors are particularly relevant in the etiology of psychiatric disorders [244]. Somewhat contrary to the beneficial effects of calorie restriction listed above, severe food restriction during pregnancy can lead to serious repercussions for the health outcomes of the offspring. Thus, individuals prenatally exposed to famine during the Dutch Hunger Winter had reduced glucose tolerance and increased risk for metabolic syndrome and showed less DNA methylation of the imprinted insulin-like growth factor 2 (IGF2) gene even six decades later [245,246]. Moreover, both the Dutch Hunger Winter and the Great Leap Forward of China, which also involved mass starvation, indicated that exposure to famine during the first trimester of gestation is associated with a higher incidence of schizophrenia and depression in adulthood [247,248].

On the other hand, a maternal high-calorie diet is also associated with the development of both metabolic and psychiatric diseases in offspring through epigenetic mechanisms [198,201,202,203,246,249,250,251]. Additionally, the choice and amounts of nutrients are important for the epigenetic milieu. Polyunsaturated fatty acids (PUFAs) that are very high in Western diets are shown to have epigenetic effects per se [252,253]. Particularly strong effects are seen during pregnancy. Mouse pups from dams treated with an omega-6 PUFA-rich diet prior to and throughout pregnancy showed disrupted cortical lamination and anxiety-like behavior, even when the omega-6 PUFA-rich diet contained a recommended omega-3-to-omega-6 ratio. Omega-6 PUFA enrichment reduced chromatin accessibility in general, but particularly of TFs downstream of cannabinoid receptor 1 (CB1), for example, that of signal transducer and activator of transcription 3 (STAT3), a factor necessary for inflammatory signaling [254]. Since omega-6 PUFAs are precursors to arachidonic acid, which in turn is a precursor to endocannabinoids, omega-6 PUFAs and, therefore, an endocannabinoid-rich diet were proposed to downregulate CB1 receptors in the offspring of the diet-treated dams via de-sensitization. It was hypothesized that de-sensitized CB1 receptors would downregulate several TFs through methylation and eventually downregulate immunoglobulin cell adhesion molecules, which alter brain development and lead to permanent neuroanatomical and psychopathological changes in the adult offspring of the omega-6 PUFA-treated dams. Dietary excesses before conception and during pregnancy thus have the potential to program the wiring of brain circuits through epigenetic mechanisms and affect emotional processing and cognition for life [254].

Several dietary factors exert epigenetic effects and modulate psychopathological conditions independent of the fatty acid or caloric content of food. The dietary factor acetyl-L-carnitine [235], which is also naturally produced by the body, increases global histone acetylation by donating its acetyl groups. This is associated with neuroprotective effects and decreased neuroinflammatory responses [255,256,257]. Another class of important dietary components, polyamines, is also suggested to exert their action by inhibiting HAT activity [258]. Polyphenols, a large family of natural compounds from plant foods present in cruciferous vegetables, soy products, and green tea, have been shown to modify the activity of DNMTis, HATis, and HDACis [259,260,261,262]. Polyphenols include resveratrol and curcumin that are well known for their antiaging properties and are considered potential adjuvant treatments for mental and brain health. However, there are mixed clinical results, potentially owing to interindividual differences and antioxidant effects of those compounds [263]. Of note, maternal resveratrol supplementation of high-calorie diets during pregnancy was found to have some benefits for maternal and fetal metabolic health in several animal studies. Resveratrol was therefore proposed to mimic the positive effects of calorie restriction and physical activity in the case of maternal overeating, which might be otherwise difficult to address while avoiding malnutrition during pregnancy [246]. It should be noted that biologically active substances of plant and animal origin are often used as prototypes or as precursors for the synthesis of new compounds with epigenetic activity [264,265]. 

#### 3.6.3. The Role of Physical Activity in Shaping the Epigenetic Landscape

Exercise increases the activity of histone acetyltransferases and histone deacetylases, reduces the level of global DNA methylation, and thus sets back the “time” of the epigenetic clock [266,267,268,269,270,271]. The changes in the epigenetic landscape go along with the therapeutic efficacy of physical exercises, which exert beneficial effects on cognitive impairment during aging, and can be used as an add-on therapy for neurodegenerative diseases [272,273] and, with less evidence, for psychopathological conditions [238,274].

### 3.7. Other Molecular Targets of Epigenetic Therapy for CNS Disorders

#### 3.7.1. TET Proteins

TET proteins perform hydroxylation of DNA at 5-methylcytosine (5mC) [275], therefore changing the probability of DNA methylation and demethylation [276]. TET proteins are involved in all aspects of neuronal development and neuroplasticity; therefore, their functional activity seems to be a significant factor in the pathogenesis of neurodegenerative and psychopathological disorders [277]. Unlike cells of other lineages, which lose the expression of TET proteins during development and differentiation, a high level of expression of TET proteins is retained in fully differentiated neurons. The TET protein expression is believed to be caused by a high level of 5-hydroxymethylcytosine (5hmC) [278] and high transcriptional competence of DNA. The neuron-specific TET expression is a promising feature that allows one to consider TET proteins to develop an accurate epigenetic intervention specific to neurons [279,280].

#### 3.7.2. Non-Coding RNAs

The most important regulators of the epigenome are non-coding RNAs. They are mostly represented by microRNAs (miRNAs), or enhancer and long non-coding RNAs (eRNAs and lncRNAs) which show a complex tertiary structure. MicroRNAs control transcription and intranuclear and cytoplasmic processing of genetic information; enhancer RNAs mainly mediate changes in the structural organization of DNA [281]. There is a potential for the use of miRNAs as antitumor agents and regenerative agents; but, at present, many drugs are withdrawn from clinical trials in the early phases due to the high frequency of side effects [282]. Currently, the possibility of using miRNAs in the treatment of neurological diseases and psychopathological conditions based on their impact on neuroimmune mechanisms is being questioned [283].

There is ample evidence of significant associations between specific miRNA expression profiles and psychopathological conditions [284,285,286,287,288,289]. However, the use of miRNAs as therapeutic agents is extremely difficult. Firstly, the spectrum of miRNA effects is wide, and it is not limited to the effect in the CNS, which increases the likelihood of side effects. This could be overcome with cell-specific delivery of miRNAs (e.g., with antibody- and nanobody-based delivery). Secondly, miRNAs are readily degraded, and there is a need to optimize the methods for their targeted organ delivery [290], for example, using “minicells” [291].

### 3.8. Clinical Trials

To assess the current state of clinical trials for substances with epigenetic activity in the treatment of pathological conditions of the CNS, an analysis of the US National Institutes of Health database (clinicaltrials.gov) [292] was carried out. Table 4 presents data on the number of active + completed (A + C) and terminated + withdrawn (T + W) studies of the compounds with primary epigenetic activity that are approved for use. The largest number of trials was conducted to assess the safety or therapeutic activity of compounds in the treatment of brain cancers, mainly of glial origin. This bias is primarily due to the exploitation of the cytostatic and cytotoxic properties of the compounds, which are especially pronounced in glial cells. 

Upon analysis, the breadth of the goals of clinical trials with valproic acid or its salts used as monotherapy or as a component of complex therapy was noteworthy. In addition to the most common pathologies targeted with valproic acid (epilepsy, affective disorders, and schizophrenia), the therapeutic efficacy of this drug is being studied in the treatment of pain syndromes, depression and PTSD, Alzheimer’s disease (one study), spinal muscular atrophy (six studies), and oncological diseases of the brain (glioblastomas, gliomas of different localizations I-IV degrees of malignancy, gliomatosis, neuroblastoma, and medulloblastoma). Hydralazine is being studied as a primary therapy for Alzheimer’s disease in one database-registered clinical trial [293]. Its effect on autophagy is seen as the leading mechanism of action. The rationale for the study of phenelzine and tranylcypromine in the treatment of depression is based on their MAO inhibitory action [294]; epigenetic effects are not considered.

## 4. Conclusions

It is well accepted that epigenetic mechanisms are the core links between pathological environmental factors and disease. In analogy, it is true that interfering with the epigenetic mechanism might be the way to prevent the development or slow down the progress of neurological and psychiatric diseases. Therefore, epigenetic therapy of diseases of the CNS appears to be a promising field in pharmacology.

The development of epigenetic therapy for CNS diseases nonetheless faces several challenges:Need for inhibitors and activators specific to different types and isoforms of epigenetic enzymes that also have good BBB permeability;Discovery of compounds with bimodal or complex activity;Understanding the changes in genome–epigenome interactions;Expansion of the bioinformatical annotation epigenetic markers at different biological levels;Evaluation of the applicability of the CRISPRon and CRISPRoff tools in humans;Further development and optimization of the cell-specific delivery of compounds or genetic constructs using viral vectors, anti- or nanobodies, or PEGylated immunoliposomes.

The expanding amount of data from studies interrogating the mechanisms of action and effects of epigenetically active drugs and compounds in the CNS strongly suggests a potential for the therapeutic use of epidrugs in brain cancer. The present review particularly outlines the prospects of using HDACis, DNMTis, and BETis for the therapy of brain neoplasms (mainly of glial origin). Analysis of the CNS-specific targets and effects of epidrugs with respect to functional regulation of brain activity reveals that they also often target glial elements, affecting microglia differentiation and secretory activity. Glia provide support, protection, and, at the same time, foster neuroinflammation processes, thus regulating them negatively and positively. Although the epigenetic biomarkers of neuroinflammation in neurological (and psychiatric) disorders are not comprehensively described [295,296,297] to give us exact targets for drugs, the strategy of epigenetic therapeutic intervention can evolve based on the glia-interfering effects of epidrugs. Fundamentally, epidrugs may control neuroinflammation, the mechanism that is at the core of many neurological and psychiatric disorders, and regulate the expression and post-translational modifications of pathological proteins. Therefore, epidrugs can be generally considered symptomatic therapies with a wide spectrum of action. Additionally, it seems reasonable to make use of complex and balanced physiological modulators of the epigenome, such as single nutrients, physical activity, and a low-calorie diet, as an add-on therapy for psychopathological and neurodegenerative conditions and age-associated dementia.

## Figures and Tables

**Figure 1 cells-12-01464-f001:**
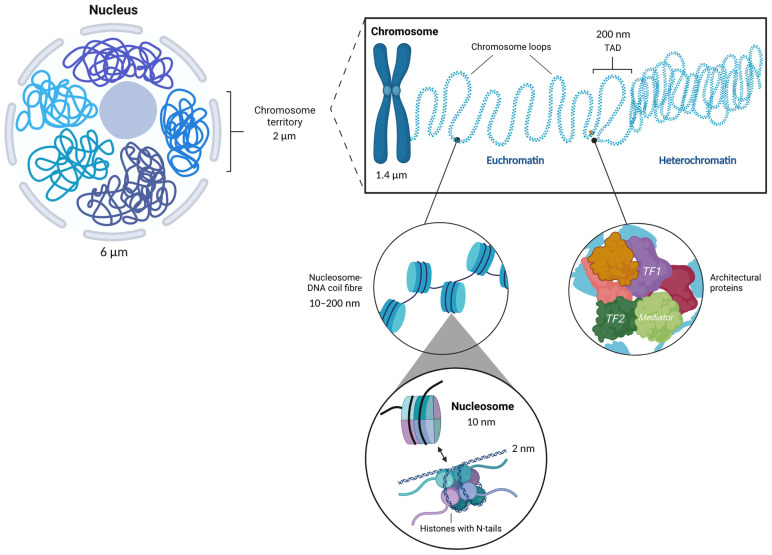
Chromatin structure. In eukaryotes, DNA (deoxyribonucleic acid) molecules in the cell nucleus are packed into chromosomes; the integral substance of which is called chromatin. Back in 1928, Emil Heitz noticed that some chromatin regions (chromosome territories) are more compact than others and described the chromosomal substance as unfolded eu- and compact heterochromatin [1]. Loosely coiled chromatin contains transcriptionally accessible DNA regions, whereas tightly coiled chromatin comprises transcriptionally inactive DNA regions [2]. One long DNA molecule is wound around globules of histone proteins (like beads on a string) and folded into a compact structure along with other proteins (architectural) and RNA (ribonucleic acid) molecules. Negatively charged 147 base pairs piece of the DNA molecule, which is wrapped around a positively charged octamer of a histone protein core [3], called a nucleosome. Nucleosomes are arranged regularly: each of the two molecules of histones 2A (H2A), 2B (H2B), 3 (H3), and 4 (H4) is folded into a compact structure in such a way that a tetramer from histones H3 and H4 is “sandwiched” between two dimers of H2A and H2B. Such packaging allows the DNA molecule, 2 m long, to be packed into a nucleus of several micrometers in size, and to be effectively subjected to the complex processes of gene reading and expression. Created with BioRender.com (accessed on 2 August 2022).

**Figure 3 cells-12-01464-f003:**
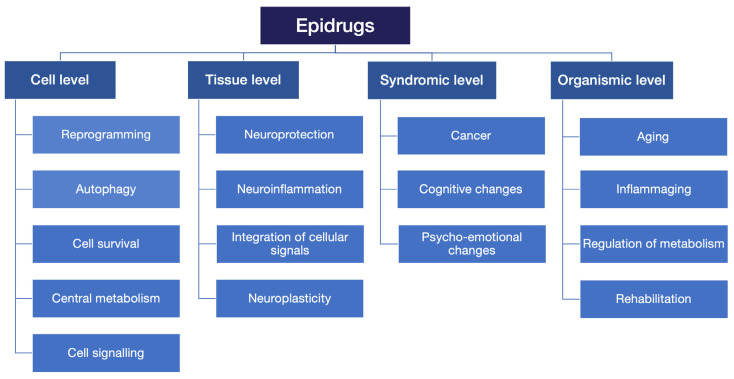
Diversity of epidrug action. Epidrugs interfere with fundamental cellular processes and provide effective maintenance of cells: on the tissue level, they affect complex phenomena of neuroprotection, neuroinflammation, and cellular integration (including neuroplasticity); systemically, they affect not only developmental and neoplastic processes but also complex biological programs. Such a spectrum of target functions suggests antiaging, anti-inflammatory, metabolic, and rehabilitative effects of epidrugs.

**Table 1 cells-12-01464-t001:** Analytical methods of epigenomics.

Epigenetic Phenomenon	Targets	Assay Types	Analytic Methods	Resolutions
DNA methylation	Methylome analysis or determination of DNA methylation/demethylation levels in CpG DNA islands	Microarray-based	Digestion-based (CHARM, DMH, MCAM, MDRE, MSRE, and HELP)	Bulk, regional, gene-specific
			Affinity enrichment (MBD and MeDIP)	Bulk, regional
			Bisulphite conversion (HumanMethylation and EPICBeadChip)	Bulk, single CpG
		Sequencing-based	Digestion-based (HELP-NGS)	Bulk, multiple genomic regions
			Affinity enrichment (MeDIP and MIRA-seq)	Bulk, regional
			Bisulphite conversion (BS-seq, BSPP, OxBS-seq, RRBS, TAB-seq, Truseq EPIC, and sWGBS)	Bulk, single-cell, genome-wide, multiple genomic regions
Chromatin accessibility and histone modifications	Chromatin proteomic profiling technologies allow for investigation of DNA–protein interactions, chromatin accessibility analysis, nucleosome positioning (nucleosome mapping), transcription factor (TF) binding analysis, etc.	Sequencing-based	Affinity enrichment (ChIP-seq and its variations, CUT&RUN, CUT&TAG, and DamID)	Bulk, single-cell, multiple genomic regions
			Open chromatin digestion-based (DNase-seq and ATAC-seq)	Bulk, single-cell, multiple genomic regions
Nuclear organization	Methods of chromosome conformation capture and genome 3D structure analysis, including spatial interactions within topologically associating domains (TADs), the activity of polycomb-group proteins and helicases, and the assessment of the architecture of the cell nucleus	Chromatin conformation capture	3C/4C/5C, Hi-C, ChIA-PET, HiChIP, and PLAC-seq	Bulk, single-cell, genome-wide, multiple genomic regions
Regulatory RNAs and epigenetic enzymes	Transcriptome analysis with RNA sequencing and expression profiling of non-coding ribonucleic acids (ncRNAs), such as microRNA (miRNA) and long non-coding RNA (lncRNA), which regulate transcription and translation; molecular biological assessment of expression and activities of the epigenetic enzymes	Sequencing- and multi-OMICS-based	RNA-seq, RIP-seq, NOMe-seq, and EpiMethylTag	Bulk, single-cell, multiple genomic regions

**Table 3 cells-12-01464-t003:** Fields of epidrug studies with respect to their effects in the CNS.

Epidrug Cluster	Instances	Some PK/PD Features	Representative Preclinical and Clinical Studies
First-generation DNMTis	5-azacytidine 5-aza-dioxycytidine	antimetabolites	potentiation of neurotoxic effects in the culture of dopamine neurons [43]
Second-generation DNMTis	hydralazine zebularine RG108 procainamide	selective inhibition of DNMT isoforms and inhibition of DNMTs and cytidine deaminase	interference with β-amyloid production [44]; enhancement of neurotoxicity [45]; and suppression of apoptosis in motor neurons [46]
First-generation HDACis	trichostatin A vorinostat (SAHA) romidepsin	reversible binding to Zn^2+^ in the HDAC catalytic center selective for HDAC I, II class enzymes	prevention of stress-related behavioral changes in mice [47,48]
Second-generation HDACis	belinostat panobinostat benzamides (e.g., MS-275) carboxylic acid derivatives (e.g., valproic acid)	improved bioavailability and less toxicity	adjuvant therapy for glioma [49]; prevention of stress-related behavioral changes in mice, antidepressant effect [50,51]; and reduction in anxiety and panic attacks in humans [52,53,54]
HATis	exifone	genome stabilization	neuroprotection [55]
HMTis	tazemetostat JNJ-64619178	duality of action: the effect is determined by the position of the methylated lysine	therapy for glioma [56]
LSDis	tranylcypromine ORY-1001	FAD-dependent inhibition (similar to inhibition of homologous LSD1 and LSD2 monoamine oxidase MAO)	change in *Bdnf* transcription, antidepressant effect in mice [57]; and alteration of *Bdnf* transcription and memory suppression [58,59]
BETis	RVX-208 JQ-1 diazepine derivatives	high-affinity BET inhibitors (antitumor and expression-regulating ApoA1 and HDL activity)	positive effect on neurogenesis in vitro, modulation of memory and learning mechanisms in mice [60,61]; and improving cognitive performance in humans [62]

**Table 4 cells-12-01464-t004:** Clinical trials of epidrugs with primary epigenetic activity for treating CNS diseases. A + C represents the number of drug-respective trials which are active or completed; T + W represents the number of drug-respective trials which are terminated + withdrawn (January 2023); *—Evaluation of the safety of chronic use of vorinostat at a dose of 100–400 mg per day for the treatment of Alzheimer’s disease.

Compound	Brain Cancer		Neurodegenerative Disorders		Psychopathological Syndromes		Others (Epilepsy; Pain Therapy)	
	A + C	T + W	A + C	T + W	A + C	T + W	A + C	T + W
5-azacytidine	3	0	0	0	0	0	0	0
belinostat	1	0	0	0	0	0	0	0
vorinostat	19	3	1 *	0	0	0	0	0
decitabine	0	1	0	0	0	0	0	0
panobinostat	5	3	0	0	0	0	0	0
romidepsin	1	0	0	0	0	0	0	0

## Data Availability

Not applicable.

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
