# Peer review of "Epidrugs in the Therapy of Central Nervous System Disorders: A Way to Drive on?"

_cells, 2023, doi:10.3390/cells12111464_

Round 1

Reviewer 1 Report

In this review, Gladkova et al., discuss the use of popular epigenetic drugs in the treatment of various brain disorders. I find that the scope of this article is far too broad to discuss anything in enough detail so a significant contribution to the field is not present. I suggest that the authors narrow the focus of the review to one of their main themes e.g. brain cancers, neurodegenerative disease, neuropsychiatric disorders, or inflammation and epigenetic therapies. 

Due to the very broad scope of this review, I have the following more specific criticisms:

·      Introduction is not cohesive. Likely because the scope of the article is so broad, I was left wondering what the review is going to cover. 

·      Sections 1 and 2  nothing is discussed in much detail

·      Very few experimental details are given. 

·      Oversimplification of data. E.g. “several of the most prevalent neurodegenerative diseases, such as Huntington’s 320 disease, Parkinson's disease, Alzheimer's disease, and amyotrophic lateral sclerosis, are 321 characterized by a global decrease in HAT activity in the brain.” There is no reference for this statement and some studies show the opposite pattern. 

·      Organization could be improved. The specific therapy for neurodegenerative diseases section jumps between AD and Huntington’s. Similarly, the section on  Psychopathological Conditions jumps between psychiatric conditions and obesity. 

·      The authors repeatedly bring up environmental factors like lifestyle and exercise but this is not discussed in much detail. 

·      There is a lot of discussion about inflammatory changes but this information is not really pulled together into a cohesive story. 

Reviewer 2 Report

In the review by Gladkova and colleagues entitled “

 Epidrugs in the therapy of central nervous system disorders: a way to drive on?”, the authors provide a review about the possibility of intervening brain disorders by using drugs that target the epigenetic landscape (epidrugs). 

The review covers a timely topic and is well-written. I have a few general suggestions for improving the manuscript:

1.    To enhance the utility of the review, I suggest to include all approved compounds that target epigenetic modifications. Table 2 gives examples of approved drugs, but I think it would be valuable to the community to have include all known drugs listed.

2.    Table 4 should be more thoroughly labelled and explained what is being shown.

3.    The review is text-heavy and thus it would be helpful to include more Figures to aid the reader. There is a lot of information here and having figures and models will significantly help.  I recommend one Figure for each of the longer subsections of section 3 (e.g. 3.4.2 and 3.4.3), but the more figures the better.

Round 2

Reviewer 2 Report

I appreciate the time that the authors put in to revising the manuscript. Most of my concerns were addressed and feel that the review is appropriate for publication.